# The Hidden Potential of High-Throughput RNA-Seq Re-Analysis, a Case Study for DHDPS, Key Enzyme of the Aspartate-Derived Lysine Biosynthesis Pathway and Its Role in Abiotic and Biotic Stress Responses in Soybean

**DOI:** 10.3390/plants11131762

**Published:** 2022-07-01

**Authors:** Raphaël Kiekens, Ramon de Koning, Mary Esther Muyoka Toili, Geert Angenon

**Affiliations:** 1Research Group Plant Genetics, Vrije Universiteit Brussel, 1050 Brussels, Belgium; raphael.kiekens@vub.be (R.K.); ramon.de.koning@vub.be (R.d.K.); mary.esther.muyoka.toili@vub.be (M.E.M.T.); 2Department of Horticulture and Food Security, School of Agriculture and Environmental Sciences, College of Agriculture and Natural Resources, Jomo Kenyatta University of Agriculture and Technology, Nairobi P.O. Box 62000-00200, Kenya

**Keywords:** DHDPS, lysine, soybean, RNA-seq, bioinformatics, abiotic, biotic, stress

## Abstract

DHDPS is a key enzyme in the aspartate-derived lysine biosynthesis pathway and an evident object of study for biofortification strategies in plants. DHDPS isoforms with novel regulatory properties in *Medicago truncatula* were demonstrated earlier and hypothesized to be involved in abiotic and biotic stress responses. Here, we present a phylogenetic analysis of the *DHPDS* gene family in land plants which establishes the existence of a legume-specific class of DHDPS, termed DHDPS B-type, distinguishable from the DHDPS A-type commonly present in all land plants. The *G. max* genome comprises two A-type *DHDPS* genes (*Gm.DHDPS-A1*; *Glyma.09G268200*, *Gm.DHDPS-A2*; *Glyma.18G221700*) and one B-type (*Gm.DHDPS-B*; *Glyma.03G022300*). To further investigate the expression pattern of the *G. max* DHDPS isozymes in different plant tissues and under various stress conditions, 461 RNA-seq experiments were exploited and re-analyzed covering two expression atlases, 13 abiotic and 5 biotic stress studies. *Gm.DHDPS-B* is seen almost exclusively expressed in roots and nodules in addition to old cotyledons or senescent leaves while both *DHDPS* A-types are expressed constitutively in all tissues analyzed with the highest expression in mature seeds. Furthermore, *Gm.DHDPS-B* expression is significantly upregulated in some but not all stress responses including salt stress, flooding, ethylene or infection with *Phytophthora sojae* and coincides with downregulation of *DHDPS* A-types. In conclusion, we demonstrate the potential of an in-depth RNA-seq re-analysis for the guidance of future experiments and to expand on current knowledge.

## 1. Introduction

L-Lysine (Lys) is one of the essential amino acids that humans and monogastric animals must acquire from the diet. In plants, Lys is biosynthesized in plastids, from the common precursor aspartate (Asp) along with three other essential amino acids: methionine (Met), threonine (Thr) and isoleucine (Ile) (Figure 1) [1]. The first and therefore considered key enzyme of the lysine-specific branch of the aspartate-derived amino acid pathway is 4-hydroxy-2,3,4,5-tetrahydrodipicolinate synthase, more commonly known as dihydrodipicolinate synthase (DHDPS, EC 4.3.3.7). DHDPS is a pyruvate-dependent class I aldolase catalyzing the conversion of pyruvate and L-aspartate-β-semialdehyde (ASA) into (2S,4S)-4-hydroxy-2,3,4,5-tetrahydrodipicolinate (HTPA). HTPA is converted into Lys by four enzymatic reactions as executed consecutively by dihydrodipicolinate reductase (DHDPR, EC 1.17.1.8), LL-diaminopimelate aminotransferase (LL-DAP-AT, EC 2.6.1.83), diaminopimelate epimerase (DAPE, EC 5.1.1.7) and diaminopimelate decarboxylase (DAPDC, EC 4.1.1.20). In addition, Lys directly regulates its own biosynthesis by inhibiting DHDPS in an allosteric way [2]. After its biosynthesis, Lys can also be catabolized resulting in the production of glutamate when it is combined with α-ketoglutarate into saccharopine and subsequently converted into allysine, as regulated in a two-step mechanism by the bifunctional lysine-ketoglutarate reductase/saccharopine dehydrogenase enzyme (LKR/SDH, EC 1.5.1.8/1.5.1.9) [3,4]. Glutamate is an important stress sensor in plants and is converted into several stress-related molecules such as proline and γ-aminobutyric acid (GABA) [5]. It recently came to attention that Lys can also act in a second catabolic pathway as a precursor for metabolites involved in the systemic acquired resistance (SAR) response of the plant [6]. In this catabolic pathway, Lys is converted into Δ1-piperideine-2-carboxylic acid (P2C) by AGD2-LIKE DEFENSE RESPONSE PROTEIN1 (ALD1) and subsequently into pipecolic acid (Pip) by SAR DEFICIENT 4 (SARD4) which in turn leads to N-hydroxypipecolic acid (NHP) biosynthesis by FLAVIN-DEPENDENT MONOOXYGENASE1 (FMO1) [7,8,9]. It has been shown that Pip in complex with FMO1 is an important immune regulatory signaling unit, orchestrating salicylic acid mediated pathogen responses, thereby directly linking Lys catabolism and the plant immune response system [10].

The existence of DHDPS in higher plants was first reported in 1975, by measuring its catalytic activity in *Zea mays*, after which studies followed in *Triticum aestivum*, *Spinacia oleracea*, *Solanum tuberosum*, *Phaseolus vulgaris*, *Pisum sativum*, *Glycine max*, *Arabidopsis thaliana*, *Coix lacryma-jobi*, *Zizania latifola*, *Vitis vinifera* and *Medicago truncatula* [11,12,13,14,15,16,17,18,19]. In all studies, the purified plant DHDPS showed high sensitivity to Lys feedback inhibition with IC_50_ values ranging from 10 to 50 µM depending on the species. In addition, protein sequence alignments and structural analysis by X-ray crystallography in *N. sylvestris*, *V. vinifera* and *A. thaliana*, shed light on conserved sites important for the catalytic activity and allosteric inhibition by Lys, as well as on the quaternary structure, with a typical ‘back-to-back’ dimer of dimers tetrameric form in plants, different from the ‘head-to-head’ confirmation in bacterial DHDPS [20,21,22,23]. In further detail, Thr68, Tyr131 and Tyr155 (*N. sylvestris* numbering is used in this paper [20]) form together the so-called catalytic triad which serves to shuttle protons, essential for part of the enzymatic reaction with stabilizing support from Thr69 and Tyr130 [20,24]. Regarding the allosteric inhibition of the DHDPS enzyme by Lys, in *N. sylvestris,* the α-amino group of Lys is known to interact with Gln73, Asn104 and Glu108; the Lys ε-amino group with Trp77, His80 and G102, and the Lys carboxyl group with Tyr130 [20]. It is interesting to note that the allosteric mechanism uses Trp77 to ‘trap’ the inhibitory Lys by closing the binding pocket as the sidechain of Trp77 flips, additionally altering the molecular dynamics in the catalytic triad due to a repositioning of Tyr130 [21,22].

Before the advent of fully annotated and publicly available genomic sequences, it was already clear that DHDPS protein sequences were highly conserved as seen when comparing *Z. mays*, *T. aestivum*, *A. thaliana*, *Nicotiana tabacum*, *Populus deltoides × P. trichocarpa* and *G. max* protein sequences [16]. Therefore, it came somewhat as a surprise that in the model legume *M. truncatula*, two DHDPS isozymes were found with multiple amino acid substitutions on positions known to be important for activity or allosteric inhibition by Lys [19]. More in detail, MtDHDPS2 showed very low expression in all tissues examined and was insensitive to Lys inhibition when expressed in *E. coli*, a feature that had only been seen until then in some bacterial DHDPS [25]. In the same study, MtDHDPS3 exhibited marginal levels of enzymatic activity when expressed in *E. coli* and was seen mainly expressed in roots and immature seeds in vivo. In addition, overexpression of MtDHDPS3 in *A. thaliana* resulted in plants with a significant increase of free Thr but not Lys in the leaves as compared to wild-type plants [19]. The novel regulatory properties of these *M. truncatula* DHDPS isozymes could have an important role in stress responses as a substitution of Gly204 to Glu204 of AtDHDPS2 in the *rsp2 A. thaliana* mutant resulted in plants with an increased resistance to oomycete *Hyaloperonospora arabidopsidis*, coupled to higher Thr levels in the leaves [26]. Additionally, analysis of microarray data shows that MtDHDPS3 is upregulated as a reaction to salt stress and upon infection with root pathogenic fungi *Phymatotrichopsis omnivore* or *Macrophomina phaseolina* [27,28,29].

Based on the findings in model legume *M. truncatula*, we want to investigate if one or more DHDPS isozymes with potentially novel molecular properties are present in the Fabaceae family in general and in the economically important legume *G. max* in particular. In addition, it is of interest to map *DHDPS* gene expression in different parts of the plant at different growth stages and to what extent *DHDPS* expression is differentially regulated upon various stress responses in *G. max*. Therefore 461 RNA-seq experiments were re-analyzed covering two expression atlases, 13 abiotic and 5 biotic RNA-seq studies of which main results are presented and discussed in this paper.

## 2. Results

### 2.1. Phylogenetic Analysis of DHDPS in Plants

To gain a better understanding of the DHDPS phylogeny and evolution in (land) plants, a total of 44 DHDPS complete protein sequences were identified in 17 plant species spanning from bryophytes to angiosperms including 6 legumes (Appendix A). The included legumes are *L. japonicus*, *M. truncatula, P. sativum*, *P. vulgaris*, *V. unguiculata* and *G. max*. More in detail, a specific DHDPS protein profile (PFAM profile hidden Markov model) was used rather than a ‘standard’ BLAST search analysis. After the identification, subsequent phylogenetic analysis of the 44 DHDPS protein sequences revealed two DHDPS clades, dividing the phylogenetic tree into what we define as the DHDPS A-type, commonly found in all land plants and the DHDPS B-type, which is legume-exclusive (Figure 2). Furthermore, each plant species in the analysis has at least one DHDPS A-type and each legume has in addition at least one DHDPS B-type. For *G. max* in particular, two DHDPS A-types (Gm.DHDPS-A1; *Glyma.09G268200*, Gm.DHDPS-A2; *Glyma.18G221700*) and one DHDPS B-type (Gm.DHDPS-B; *Glyma.03G022300*) are present [19].

The existence of two DHDPS subtypes in (land) plants is also reflected in the amino acid substitution table for sites known to be important in catalytic activity or allosteric inhibition by Lys (Table 1). It is clear that amino acids are much more conserved within the DHDPS A-type subgroup as compared to the DHDPS B-type subgroup. In the DHDPS A-type sequences, an Ile222 is substituted by Val222 or Met222 and Gln73 is being replaced by His73 while Tyr130, Tyr131, Tyr155, Arg160, Lys183 and Gly102, Asn104 are conserved for all DHDPS. For the DHDPS B-type sequences, the amino acid substitution pattern is more diverse as compared to the DHDPS A-type sequences. Interesting amino acid substitutions within the DHDPS B-type include a Thr68 to Ser68 which is an amino acid position from the ‘catalytic triad’ and a substitution of the DHDPS A-type conserved His80 into Gln80 or Lys80 in the Lys allosteric binding site. When focusing on *G. max* DHDPS, both Gm.DHDPS-A1 and Gm.DHDPS-A2 follow the consensus sequence of the table with one exception being Ile222 substituted for Val222 as also seen in *A. coerulea*, *P. trichocarpa* and *P. sativum* (Table 1). The only DHDPS B-type, Gm.DHDPS-B, has amino acid substitutions His80Gln, His111Lys, Asp207Lys, Ile222Gln and Asn261Val. Remarkably, there is no amino acid substitution in Gm.DHDPS-B at Thr68 from the ‘catalytic triad’ as seen in some other B-types from *L. japonicus*, *M. truncatula*, *P. sativum*, *P. vulgaris* and *V. unguiculata*. It is interesting to note that Eg.DHDPS-A2, -A3, and -A4 from *E. grandis* show amino acid substitutions not seen in DHDPS A-type nor DHDPS B-type sequences which is also reflected in the phylogenetic tree in which these *E. grandis* sequences form a small subclade within the DHPDS A-type clade.

### 2.2. RNA-Seq Re-Analysis Method Validation

A bioinformatics pipeline was created for re-analysis of RNA-seq raw data files in order to filter out expression data for any given gene—or set of genes of interest, being in our case *DHDPS* in *G. max*. The pipeline starts with downloading raw FASTQ files from the database and subsequently the reads per file are trimmed on quality, mapped to the reference genome, counted per gene and put in a human-readable count matrix. The gene expression count files can then be used for differential expression analysis or for calculating the RPKM (Reads Per Kilobase of transcript per Million reads mapped) values within each sample (Figure 3). As a pilot study, the bioinformatics pipeline was validated by comparing a publicly available gene atlas at https://www.soybase.org (accessed on 1 December 2020) with the re-analysis of the raw FASTQ files [31]. For this experiment, 14 plant samples were originally used with one biological repeat per sample. The re-analysis showed a high and significant correlation (*p* < 0.0001) with the original raw expression data for all samples and mapped up to 72% more unique reads in comparison with the original read mapping (Table 2). Correlation plots of this analysis can be found in Appendix A.

Next, the log2-transformed RPKM data were filtered out for the three *DHDPS* genes in *G. max* in the original and the re-analyzed dataset, showing a very comparable expression pattern, validating the pipeline at the gene level (Figure 4). Despite no statistical test can be performed when having only one biological repeat per sample, we do see that *Gm.DHDPS-A1* and *Gm.DHDPS-A2* have very similar expression levels within each sample and *Gm.DHDPS-B* is rarely expressed except in the root and the nodule (Figure 4). Both *Gm.DHDPS-A1* and *Gm.DHDPS-A2* are highly expressed in young leaves, but also in most seed developmental stages and seed pods while they are expressed at a relatively lower level in the root, nodule and flower samples. Remarkably, low but observable expression (RPKM < 0.4) of *Gm.DHDPS-B* is found in the pod and early seed developmental stages. These low RPKM values were not picked up in the original data analysis due to the rounding of values below 1 to 0 [31].

### 2.3. Novel DHDPS Expression Atlas Data by RNA-Seq Re-Analysis

After validation of the re-analysis pipeline, knowledge of the *DHDPS* expression pattern in *G. max* was expanded by using the dataset from Yanting Shen and co-workers [32]. This dataset from study ‘SRP038111′ comprises RNA-seq data for 28 plant samples from various growth stages and was originally used for genome-wide identification of alternative splicing events in *G. max* [32]. After re-analysis, normalized and log2-transformed expression values for the *DHDPS* genes were filtered out within the 28 plant samples in *G. max* (Figure 5). Expression values for *Gm.DHDPS-A1* and *Gm.DHDPS-A2* are comparable within each sample and are relatively higher in young leaf tissue such as leaf buds in the germination stage compared to cotyledons in both the trefoil or germination stages and old or senescent leaves (Figure 5). *Gm.DHDP-A1* and *Gm.DHDPS-A2* are also relatively highly expressed in root and flower tissue in all growth stages but are higher expressed in old flowers as compared to young flowers. Interestingly, *Gm.DHDPS-A*1 and *Gm.DHDPS-A2* expression levels decrease systematically from young pods with tiny young seeds (14-DAF to 28-DAF) to older pods without the seeds (21-DAF to 35-DAF) while they increase from younger seed to older seeds (21-DAF to 45-DAF), after which it decreases dramatically in 70-DAF full-grown mature seeds (RPKM < 1). The highest *Gm-DHDPS-A1* and *GmDHDPS-A2* RPKM values are found both in 42-DAF seeds with an RPKM of 4.26 and 4.36, respectively. For *Gm.DHDPS-B,* the expression is the highest in the roots at the germination stage compared to all other samples with an RPKM value of 1.94 and is in contrast very low to not expressed at all in most other tissues at different growth stages. Interestingly, the second-highest *Gm.DHDPS-B* RPKM value of 1.31 is found in the cotyledon of the plant at the trefoil stage, but is not found to be expressed in the cotyledon from the germination stage. Also noteworthy is the relatively low but detectable *Gm.DHDPS-B* expression in the stem at the germination stage, but also in senescent leaves with RPKM values of 0.56 and 0.53, respectively.

### 2.4. The Effect of Biotic and Abiotic Stress on the Expression of DHDPS and Other Genes Involved in the Aspartate-Derived Lys Biosynthesis and Catabolic Pathway

DHDPS is the first and key enzyme in the Lys biosynthesis branch of the aspartate-derived amino acid production pathway (Figure 1). To investigate the effect of abiotic or biotic stress responses on *DHDPS* gene expression in *G. max*, 13 abiotic and 5 biotic studies were re-analyzed (Appendix A). For each experiment, the differential expression values were calculated genome-wide and subsequently filtered out for *DHDPS* together with all other genes of the Lys-biosynthesis and -catabolism pathway to get a holistic view of *DHDPS* expression in relation to the other genes of the pathway (Appendix A). Some quality control steps in terms of the expression patterns were performed by analyzing the volcano plots in addition to a PCA (Principal Component Analysis) for each experiment (Appendix A). Volcano plots for experiments SRP024277 (ozone treatment), SRP024277 (drought), SRP031889 (Fe deficiency), SRP058975 (water deficiency), SRP064384 (CO_2_ and drought) showed a relatively low amount of significant differentially expressed genes implying a smaller chance to find *DHDPS* to be significant differentially expressed in these experiments as well. In one experiment, being SRP056137 (*Fusarium oxysporum* infection), the volcano plot of the infection with the non-pathogenic strain at 96 hpi (hours post infection) is very irregular and results thereof should be omitted for further analysis. The latter experiment also showed no clear sample clustering by PCA analysis. Boxplots were used to graphically represent the variation in differential expression of all experiments per gene (Figure 6). For all three *DHDPS* genes (Figure 6, box) most of the Log2Fold change values fall within −1 and 1 which means that no noteworthy up- or downregulation can be detected. However, for *Gm.DHDPS-B* the variation for the abiotic stress experiments is larger as compared to *Gm.DHDPS-A1* and *Gm.DHDPS-A2,* with two outliers representing Log2Fold change values of 4.0 (*p* < 0.05) and 4.1 (*p* < 0.05) in leaf petioles after 24 h and 48 h of ethylene treatment, respectively [33]. Additionally, a negative outlier for *Gm.DHDPS-B* with a Log2Fold change value of −3.5 in the leaves after 1 h of salt treatment was found, however, it was not statistically significant (*p* > 0.05) [34]. Even more negative Log2Fold change values can be found for *Gm.DHDPS-B* during biotic stress after 12 h and 24 h of SMV treatment with values of −5.0 and −4.8, respectively, but both not being statistically significant (*p* > 0.05). Interestingly, there are nine abiotic stress experiments in which *Gm.DHDPS-A1* or *Gm.DHDPS-A2* show Log2Fold change values lower than −1, while no Log2Fold change values higher than 1 or lower than −1 were found for both *Gm.DHDPS-A* genes in biotic stress experiments.

For subsequent analysis, only the statistically significant (*p* < 0.05) Log2Fold change values higher than 1 or lower than −1 for *DHDPS* were filtered out, resulting in Table 3. For the experiments in which a *DHDPS* gene was statistical significantly differentially expressed, no irregular volcano plots or inconsistent sampling clustering by PCA analysis were observed (Appendix AA). The ethylene treatment experiment is well represented in Table 3, with 10 out of 21 statistically significant results [33]. In this ethylene experiment, *Gm-DHDPS-A1* or *Gm.DHDPS-A2* is, with exception of *Gm.DHDPS-A1* 12 h after ethylene treatment in the leaf abscission zone, significantly downregulated, while, in contrast, *Gm.DHDPS-B* is upregulated with a Log2Fold change of 4.0 and 4.1 after 24 h and 48 h of ethylene treatment in the leaf petiole, respectively. Furthermore, *Gm-DHDPS-B* is significantly upregulated in a flooding, water deficit, ozone and salt experiment with the exception of one salt treatment from a study in the root in which *Gm-DHDPS-B* is downregulated with a Log2Fold change of −1.7 (*p* < 0.001) [34,35,36,37,38]. Interestingly, in the experiment in which gene expression was measured 24 h after water deficit, a significant upregulation of *Gm-DHDPS-B* (Log2Fold change of 1.3, *p* < 0.05) coincides with a significant downregulation of both *Gm.DHDPS-A1* and *Gm.DHDPS-A2* with Log2Fold change values of −1.1 (*p* < 0.001) and −2.4 (*p* < 0.001), respectively [35]. Less significant differential expression for *DHDPS* was found in the biotic stress experiments as compared to the abiotic stress experiments. Yet, a significant downregulation of the *Gm.DHDPS-A1* gene was seen 4 days after infection of *P. soja* in the root with a Log2Fold change value of −1.0 (*p* < 0.05) and a significant upregulation of *Gm.DHDPS-B* in roots 15 days after infection with nematodes with a Log2Fold change value of 1.2 (*p* < 0.001) [39,40].

For the other enzymes from the Lys biosynthesis pathway, most variation of the Log2Fold changes fall between −1 and 1 with an exception for *Gm.DAPDC* (Glyma.10G053600) as it shows a clearly different expression pattern when compared to its two paralogues (Glyma.03G181200) and (Glyma.13G140700). Furthermore, *Gm.DAPDC* clearly shows downregulation during abiotic stress as opposed to upregulation during biotic stress. In addition, many negative outliers can be spotted for most genes in general during abiotic stress, but not for biotic stress. An example is the highest downregulation for *AK* (Glyma.19G102100) after 72 h of ethylene treatment in the leaf abscission zone with a Log2Fold change value of −6.7 (*p* < 0.001) [33]. It is interesting to note that genes involved in the Lys catabolism are strongly upregulated upon abiotic or biotic stress responses. For example, the highest upregulation found in the re-analysis was a Log2Fold change of 10.4 (*p* < 0.001) for *Gm.LKR/SDH* (Glyma.13G115500) in the shoot after 24 h of water deficit [35].

While differential expression analysis can be very useful and informative, as shown in Figure 6 and Table 3, it can be somewhat misleading when absolute expression values are low and small differences between control and treatment appear to be large, especially when being log2-transformed. Therefore, RPKM values were in addition plotted as line graphs for the studies in which at least one *DHDPS* was significantly up- or down regulated (*p* < 0.05), resulting in Figure 7. A clear distinct expression pattern for *Gm.DHDPS-B* is seen in general as compared to *Gm.DHDPS-A1* and *Gm.DHDPS-A2* (Figure 7). More in detail, for the ozone experiment, a downregulation for the *Gm.DHDPS-A2* gene but not for *Gm.DHDPS-A1* is noted, while a relatively small upregulation of the *Gm.DHDPS-B* gene is present, the latter, however, at a much lower expression level in the treatment compared to control (<1 RPKM) [37]. Next are two independent salt experiments with differences in experimental conditions and genotypes. In the first study, 100 mM salt and genotype Wm82 were used compared to 150 mM salt and genotype C08 [34,38]. While in both salt experiments the *Gm.DHDPS-A* expression resides between 2 and 3 RPKM, more variation in expression is seen for *Gm.DHDPS-B*. Time points for sampling are also different in both studies but in general, the *Gm.DHDPS-B* gene is downregulated in the early—and upregulated in the late salt response as compared to control. Next, for the ethylene treatment experiments, a dramatic decrease in both *Gm.DHDPS-A* gene’s expression levels towards 72 h after ethylene treatment as compared to control are noted. In contrast, *Gm.DHDPS-B* expression, albeit being relatively low in general, seems to be upregulated 24 h and 48 h after ethylene treatment, to decrease afterwards, meeting *DHDPSA* gene’s final basal expression levels [33]. Interestingly, a very similar expression pattern is observed in the air exposure experiments [35]. As seen in Table 3, the flooding experiment shows significant upregulation of *Gm.DHDPS-B* (Log2Fold change value of 2.7; *p* < 0.05), however, when looking at the line graph, this increase, although statistically significant, is relatively low compared to *Gm.DHDPS-A* gene’s expression levels [36]. In the biotic stress study, *Gm.DHDPS-B* is upregulated 4 days post-infection (dpi) with oomycete *P. sojae,* while both *Gm.DHDPS-A1* and *Gm.DHDPS-A2* show a downregulation at 4 dpi and upregulation again at 7 dpi [39]. Finally, upon treatment with nematodes in the nematode-resistant cultivar ‘Huipizhi Heidou’, all three *DHDPS* genes show a similar expression pattern for 0, 5 and 10 dpi, but at 15 dpi, *Gm.DHDPS-B* is upregulated, while both *GmDHDPS-A* genes are downregulated compared to control [40].

## 3. Discussion

*DHDPS* is known to be encoded by a small-sized gene family, as it was found to have two gene copies with high sequence similarity in *A. thaliana*, *Z. latifolia* and *T. aestivum* [17,41,42]. To investigate more in detail the *DHDPS G. max* sequences and how they map within land plant evolution, an extensive phylogenetic analysis was performed. For this, a plant-specific DHDPS profile hidden Markov model (profile HMM) was used on 11 evolutionary diverse plant species in addition to 6 legumes, including *G. max*, with high-quality genome data publicly available. There is strong evidence (100% bootstrap support; Figure 2) for *DHDPS* being biphyletic with a *DHDP-A* type common in all land plants and a *DHDPS-B* type, not found in any other plant species except legumes. This implies that the *DHDPS-B* type evolved after the first whole-genome duplication (WGD) event in the *Fabales* around 60 million years ago and was subject to positive selection within the *Fabaceae* [43]. Previous research in *M. truncatula* describes a *DHDPS* gene less sensitive to Lys inhibition (*Mt.DHDPS-B4* in this paper) and one other *DHDPS* gene with strongly decreased activity (*Mt.DHDPS-B2* this paper) [19]. Interestingly, there is strong support in the phylogenetic tree (96% bootstrap; Figure 2) for a putative ‘Lys less-sensitive’ subclade in which *Mt.DHDPS-B4* is present along with three *DHDPS* genes from *L. japonicus* and one from *P. sativum*. However, this hypothesis needs to be confirmed at the molecular level. This hypothesis would also imply that not all legumes possess a ’Lys less-sensitive’ *DHDPS* gene, including *G. max,* which has only one *DHDPS-B* type gene (*Gm.DHDPS-B*), more closely related to *Mt.DHDPS-B2* than to *Mt.DHDPS-B4*. It is possible that a spectrum of Lys feedback inhibition and differences in catalytic activity is present within the *DHDPS-B* types yet to be determined experimentally. For *Gm.DHDPS-B,* no amino acid substitution was found at position 68 from the catalytic triad, while such substitution is present in the closely related species *P. vulgaris* and *V. unguiculata* (Table 1). It is known that an identical mutation at this position in *E. coli* leads to severe impairment of the enzyme’s activity and could explain the marginal activity of *Mt.DHDPS-B2* as seen in vitro [19]. This leads to the hypothesis that *Gm.DHDPS-B* could be more enzymatically active than expected, as compared to *Mt.DHDPS-B2*. Other interesting amino acid substitutions in *Gm.DHDPS-B* are H80Q and H111K, from the allosteric inhibition site that could define other yet existing subgroups within the *DHDPS-B* type genes. It would be interesting to target specific *Gm.DHDPS-B* amino acid substitution sites for mutation analysis in vitro and in vivo.

Designing new experiments to unravel the functionality of *DHDPS* genes in *G. max* without exploring the massive amount of RNA-seq expression data that is publicly available online, would be unwise. However, finding expression data for your gene of interest can be quite cumbersome as this data is often ‘trapped’ within the RNA-seq data files. Therefore, a customized bioinformatics pipeline was built to extract expression data for *G. max DHDPS* genes along with the other genes involved in the aspartate-derived Lys biosynthesis pathway (Figure 1, Appendix A). Indeed, RNA-seq data analysis always forms a certain challenge as a sequence of specialized programs need to be used and a wide choice of these programs exists for each step in the analysis [44]. The analysis pipeline described here is relatively easy to set up, uses open-source programs only, and can be used not only for *G. max*, but for any species of interest if a full genomic sequence and qualitative gene annotation is available. It should be noted that for this pipeline, the STAR program was chosen for the mapping of the reads as it has been shown to outperform many other mapping tools and preferred DESeq2 over others for the differential expression analysis, as it is one of the safest and most precise compared to its competitors [45,46]. The pipeline was validated by comparing the publicly available SoyBase expression atlas data with the re-analyzed data of the original raw FASTQ files [31]. A high and statistically significant correlation was found between the original and re-analyzed genome-wide data and we mapped in general more unique reads as compared to the original analysis, probably due to the use of more recent gene annotation (Wm82.a2.v1 vs. Wm82.a1.v1) [31]. For *DHDPS* more specific an almost identical expression pattern was found between the SoyBase data and the re-analyzed data supporting the validity of the pipeline (Figure 4). The RNA-seq atlas data was expanded with a re-analysis of the data from a study of Shen et al. (2014) [32]. When combining both expression atlas data analyses, it is clear that *Gm.DHPDS-A1* and *Gm.DHDPS-A2* expression is almost identical in all tissues in all samples. This is in contrast to reports in *A. thaliana* in which tissue and cell specificity of each DHDPS-A type was seen [47,48], however, in agreement with the more recent high resolution Klepikova atlas in *A. thaliana,* in which both *DHDPS* genes are equally expressed in all plant samples [49]. As in the latter *A. thaliana* atlas, *Gm.DHDPS-A* is expressed to the highest degree in mature seeds and relatively less in young seeds, while the lowest expression is found in senescent leaves. *Gm.DHDPS-B* on the other hand is expressed in a tissue-specific manner. The highest *Gm.DHDPS-B* expression is seen in the root samples, but also relatively high expression is found in cotyledons at the trefoil stage yet not expressed in the cotyledon at the germination stage. This observation suggests an upregulation of *Gm.DHDPS-B* to compete with *Gm.DHDPS-A,* thereby affecting and regulating Lys biosynthesis by molecular mechanisms not yet fully understood. The *Mt.DHDPS-B2* gene closely related to *Gm.DHDPS-B* is also relatively highly expressed in the root, while being close to undetectable in leaves and immature seeds, as measured by RT-qPCR [19]. However, *Mt.DHDPS-B2* was seen to be highly expressed in mature seeds, a feature not seen for *Gm.DHDPS-B* in our analysis. This suggests that not all *DHDPS-B* type genes are transcriptionally regulated in the same manner. A promoter analysis for *Gm.DHDPS-B* could shed light on this matter.

The main objective of the research described this paper was to investigate whether *DHDPS* in *G. max* is differentially expressed in response to any abiotic or biotic stress responses and to what extent. In total, 93 RNA-seq experiments were retained for a genome-wide re-analysis and data for *DHDPS* and 25 other genes from the Lys biosynthesis and catabolic pathways were subsequently filtered out. Although we cannot compare between studies (different treatments, genotype, etc.), a meta-analysis is still useful to obtain an overview of the overall variation in gene expression (Figure 6). More specifically, *Gm.DHDPS-B* expression showed more variation as compared to *Gm.DHDPS-A1* and *Gm.DHDPS-A2* which suggests that stress signaling acts directly or indirectly on the Lys biosynthesis pathway through *Gm.DHDPS-B* rather than the *Gm.DHDPS-A* genes. Indeed, we note a significant upregulation of *Gm.DHDPS-B* after ethylene treatment in de leaf petiole, flooding in the leaf, water deficit in the shoot, ozone treatment in the leaf, water deficit in the shoot, high salt in the root and after nematode treatment in the root. Interestingly, upregulation of *Gm.DHDPS-B* upon specific abiotic stresses coincides with significant down-regulation of both *Gm.DHDPS-A1* and *Gm.DHDPS-A2* genes, suggesting opposite acting transcriptional regulation mechanisms yet to be determined. This effect is well illustrated in two separate salt stress studies where *Gm.DHDPS-B* becomes the dominant *DHDPS* type in root cells, from 6 h up to 48 h after treatment (Figure 7) [34,38]. By using a recently published online expression atlas tool in *M. truncatula* we see also an upregulation of *Mt.DHDPS-B2* while both *Mt.DHDPS-A1* and *Mt.DHDPS-A2* are downregulated in roots as a reaction to salt stress [50]. It is surprising that *Gm.DHDPS-B* is upregulated in many leaf tissues under stress, while normally not expressed in leaf tissue at all, except for old cotyledons or senescent leaves (Figure 4). Finally, there is no significant differential expression for any *DHDPS* in experiments with high or low temperature; Fe, K, N or P deficiency; elevated CO_2_ levels; Si^−^ treatment; high or low pH; infection with *Fusarium oxysporum* or soybean mosaic virus (SMV).

In this paper, we focus on *DHDPS*, but some additional interesting findings were obtained. A first example is *Gm.DAPDC,* which clearly is more responsive to abiotic and biotic stresses, as compared to its paralogues. A second example is that genes involved in Lys catabolism (via both the saccharopine and pipecolic acid pathways) are strongly upregulated upon abiotic or biotic stress. LKR/SDH is known to be strongly upregulated upon infection with *Pseudomonas syringae* and under salt or osmotic stress in *A. thaliana*, acting in mechanisms not fully understood yet. On the other hand, genes involved in the SAR response are known to be upregulated more upon biotic—rather than abiotic stresses, as seen in *A. thaliana* and confirmed by our re-analysis in *G. max* [51].

In conclusion, we demonstrate that RNA-seq re-analysis can be very useful to gain more insight in gene expression patterns of a candidate gene and can guide the researcher in the smart design of future experiments. A novel legume specific DHDPS subclade was found, distinguishable from the DHDPS type commonly found in all land plants, with one representative in *G. max*, being Gm.DHDPS-B. There is strong evidence that Gm.DHDPS-B is connected with ethylene, salt and osmotic stress, but also with infection by plant pathogen *P. sojae*. To date in *G. max*, only the enzymatic properties of Gm.DHDPS-A1 were studied, but never compared to Gm.DHDPS-B [14]. It will be interesting to explore the functional characterization of Gm.DHDPS-B both in vitro and in vivo.

## 4. Materials and Methods

### 4.1. Phylogenetic Analysis of DHDPS

A gene detection method was used similar to the one described in [52]. With ‘DHDPS’ as a query (identifier PF00701 in the PFAM database), a FASTA-formatted file was generated through the EMBL-EBI PFAM database (http://pfam.xfam.org/, accessed on 21 June 2022), using the online ‘Species Distribution’ tool and selecting for *Viridiplantae* only [53], resulting in a raw FASTA file with 191 DHDPS protein sequences from 82 different plant species. Of these 191 DHDPS plant-specific protein sequences, 35 were manually removed due to being tagged as incomplete sequences or clearly inconsistent with the DHDPS highly conserved protein consensus sequence. The remaining 156 full-length plant-DHDPS protein sequences were aligned using the *MUSCLE* V5 (Edgar RC, New York, US) multiple alignment program as implemented in the *MEGA* software (v7.0) [54,55]. From this DHDPS multiple alignment sequence, a HMMER motif was built using the *Hmmbuild* software (*HMMER* V3.1b2, Sean R. Eddy, Maryland, US) and the resulting plant-specific DHDPS-HMMER motif was subsequently used as input for the *Hmmsearch* software (*HMMER* V3.1b2, Sean R. Eddy, Maryland, US) to search for DHDPS-like sequences within the publicly available proteomes of *Physcomitrella patens (Pp)*; JGI v3.3, *Selaginella moellendorffii (Sm)*; JGI v1.0, *Picea abies (Pa)*; ConGENIE v1.0, *Oryza sativa (Os)*; MSU v7.0, *Zea mays (Zm)*; MGSP refgen v4, *Sorghum bicolor (Sb)*; JGI v3.1, *Aquilegia coerulea (Ac)*; JGI v3.1, *Vitis vinifera (Vv)*; Genoscope 12X, *Eucalyptus grandis (Eg)*; JGI v2.0, *Arabidopsis thaliana (At)*; TAIR10, *Populus trichocarpa (Pt)*; JGI v4.1, *Glycine max (Gm)*; Wm82.a2.v1, *Vigna unguiculata (Vu)*; JGI v1.2, *Phaseolus vulgaris (Pv)*; JGI v2.1, *Pisum sativum (Ps)*; INRA-Genoscope v1a, *Medicago truncatula (Mt)*; Mt4.0v1 and *Lotus japonicus (Lj)*; Gifu v1.2. Sequences with significant *Hmmsearch* hits (E-value < 1 × 10^−5^) were retained for analysis. For phylogenetic analysis, the Neighbor-Joining (NJ) algorithm was used, as implemented in the MEGA v7.0 software (Koichiro Tamura, Tokyo, Japan) package with standard settings and the number of bootstraps set to 500, on full-length protein DHDPS sequences aligned using the *MUSCLE* alignment tool within MEGA v7.0 [54,55]. In addition, an amino acid substitution table was created for DHDPS positions known to be important for catalytic activity and Lys inhibition using an amino acid color scheme depicting the physicochemical characteristics of each amino acid, as described in [30].

### 4.2. Identification of Enzymes Involved in Lysine Biosynthesis and Catabolism Other than DHDPS

To identify the soybean enzymes AK, AK/HSDH, ASADH, DHDPR, LL-DAP-AT, DAPE, DAPDC involved in the Lys biosynthesis and LKR/SDH from the Lys catabolic branch, several bioinformatic prediction tools were combined. First, a list with predicted enzymes was downloaded from the ‘L-lysine biosynthesis VI pathway’ in combination with the ‘lysine degradation pathway II’ of the PlantCyc database (v15.0) with the organism set to *G. max* [56]. This list was manually checked and the protein sequence of the first enzyme listed per enzymatic reaction (enzyme commission number) was used as a query for a pBLAST on the Phytozome website, with standard settings and *G. max* set as target species [57]. In addition, the identification of predicted soybean enzymes involved in the systemic acquired resistance response (ALD1, SARD4, FMO1) were obtained with the same pBLAST method, however, using the *Arabidopsis thaliana* protein sequences as published in their respective papers [7,8,9]. All pBLAST results were checked manually and only hits with significant results were retained for the final target enzyme list (bit score > 250).

### 4.3. RNA-Seq Data Re-Analysis and Differential Gene Expression Analysis

The query ‘(*G. max*) AND “*G. max*” [orgn:txid3847]’ was used in the search field of the publicly available sequence read archive (SRA) database of the National Center for Biotechnology Information (NCBI) (https://www.ncbi.nlm.nih.gov/sra, accessed 1 December 2020) [58]. In addition, ‘RNA’ was selected as a data source and the results were linked out to a downloadable list using the ‘SRA Run Selector’ tool. This list was filtered on ‘ILLUMINA’ as a ‘Platform’ and ‘RNA-seq’ as the ‘Assay type’ while both ‘SINGLE’ and ‘PAIRED’ library sources were retained, resulting in 2671 records. Subsequently, 2 RNA-Seq atlas data studies (SRP025919, SRP038111), 13 abiotic stress studies (SRP009826, SRP024277, SRP031889, SRP035871, SRP041622, SRP045932, SRP050050, SRP076153, SRP058975, SRP064384, SRP105922-SRP105965, SRP132150, SRP108540) and 5 biotic stress studies (SRP155375, SRP056137, SRP091708, SRP126743, SRP135932) were manually selected resulting in a final list of 461 raw FASTQ files to analyze [31,32,34,35,36,37,38,39,40,59,60,61,62,63,64,65,66,67]. An additional criterion for the selected biotic and abiotic RNA-Seq experiments was to have at least two biological repeats per treatment and control sample. Each of the 461 raw FASTQ files selected for analysis were converted into gene count files using a customized Bash shell script. This pipeline script starts with the Wget command downloading the FASTQ file from the European Nucleotide Archive (ENA) repository of the European Bioinformatics Institute (EBI), except for FASTQ files from study SRP025919 which were downloaded from the SoyBase database directly [68,69]. Next, the FASTQ file was trimmed using Sickle (v1.33) set to a length threshold of 35 bp and a quality threshold score of 20 [70]. Trimmed reads were mapped using STAR (v2.6.0) on the Wm82.a2.v1 genome assembly, except for trimmed reads from the SRP025919 study which were mapped to the Wm82.a1.v1 genome assembly, making comparison with the original data analysis possible [71]. Each resulting sorted BAM file was used by HTSeq (v0.9.1) for counting the number of sequence reads per gene [72]. HTSeq-count files were used directly or normalized using the ‘Reads Per Kilobase of transcript per Million reads mapped’ (RPKM) method by dividing the raw count of the gene by the total number of mapped reads per million and the length of the gene in kilobases. Differential expression re-analysis was performed for the 93 experiments with the DESeq2 software package (v3.12) in R studio (v1.3.1093) using the non-normalized HTSeq-count files as input and outputs a results table including the gene ID, base mean, Log2FoldChange, *p*-Value and adjusted *p*-Value per gene. For selection of significant results, corrections for multiple differential expression were taken into account by using the adjusted *p*-Value from DESeq2 [73]. As a quality control step of the DESeq2 results, a volcano plot for each experiment was generated in the DESeq2 workflow for each result using R-code, and a PCA analysis was performed using the pcaExplorer package clustering on condition (treatment and control) (v2.22.0) [74].

### 4.4. Statistical Analysis and Graphic Representations

Statistical analysis and graphics were executed in R studio (v1.3.1093). Pearson’s correlation coefficients were calculated with the cor.test function. Bar charts, scatter plots, box plots and line graphs were created with a colorblind-friendly palette using the ggplot2 software package (v3.3.3) [75].

## Figures and Tables

**Figure 1 plants-11-01762-f001:**
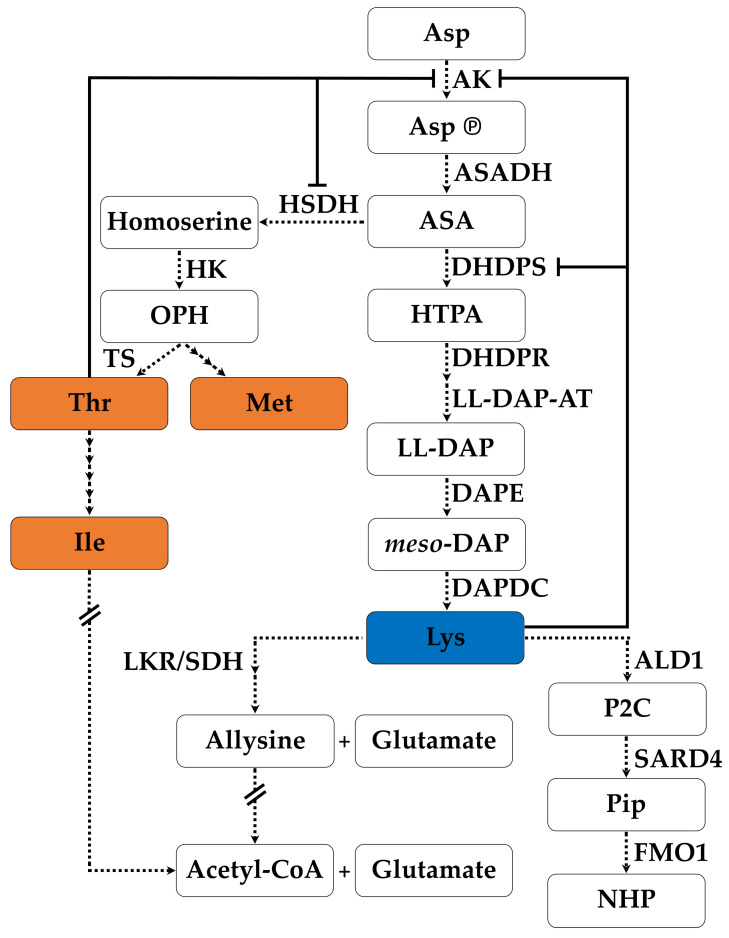
The Aspartate-derived amino acid pathway in plants branching into the biosynthesis of essential amino acids lysine (Lys), threonine (Thr), isoleucine (Ile) and methionine (Met). Lys is subject to catabolism into allysine and Acetyl-CoA or used for biosynthesis of Pipecolic acid (Pip) required for the systemic acquired resistance (SAR) response in plants. Dashed arrows represent enzymatic reactions with the generic enzyme names next to each arrow. An arrow interrupted by a double dash indicates multiple yet not further specified (bio)chemical reactions. Full lines represent negative feedback regulation. Asp: L-asapartate; Asp℗: L-aspartyl-4-phosphate; ASA: L-aspartate-β-semialdehyde; HTPA: 4-hydroxy-2,3,4,5-tetrahydrodipicolinate; LL-DAP: L,L-diaminopimelate; meso-DAP: meso-diaminopimelate; Acetyl-CoA: S-acetyl Coenzyme A; P2C: Δ1-piperideine-2-carboxylic acid; NHP: N-hydroxypipecolic acid; OPH: O-phosphohomoserine; AK: aspartate kinase; ASADH: aspartate semialdehyde dehydrogenase; DHDPS: 4-hydroxy-2,3,4,5-tetrahydrodipicolinate synthase; DHDPR: dihydrodipicolinate reductase; LL-DAP-AT: LL-diaminopimelate aminotransferase; DAPE: diaminopimelate epimerase; DAPDC: diaminopimelate decarboxylase; LKR/SDH: lysine ketoglutarate reductase/saccharopine dehydrogenase; ALD1: AGD2-LIKE DEFENSE RESPONSE PROTEIN1; SARD4: SAR DEFICIENT 4; FMO1: FLAVIN-DEPENDENT MONOOXYGENASE1; HSDH: homoserine dehydrogenase; HK: homoserine kinase; TS: threonine synthase.

**Figure 2 plants-11-01762-f002:**
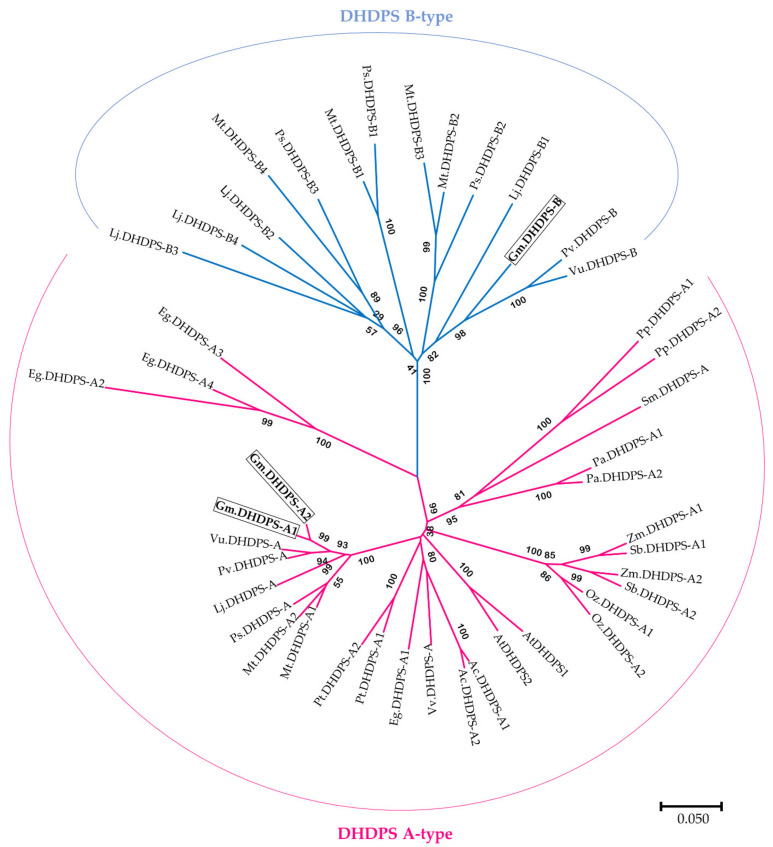
Phylogenetic NJ tree of DHDPS protein sequences of *Physcomitrella patens* (Pp), *Selaginella moellendorffii* (Sm). *Picea abies* (Pa), *Oryza sativa* (Os), *Zea mays* (Zm), *Sorghum bicolor* (Sb), *Aquilegia coerulea* (Ac), *Vitis vinifera* (Vv), *Eucalyptus grandis* (Eg), *Arabidopsis thaliana* (At), *Populus trichocarpa* (Pt), *Glycine max* (Gm), *Vigna unguiculata* (Vu), *Phaseolus vulgaris* (Pv), *Pisum sativum* (Ps), *Medicago truncatula* (Mt) and *Lotus japonicus* (Lj). The tree is drawn to scale, with branch lengths in the same units as those of the evolutionary distances used to infer the phylogenetic tree. Bootstrap support numbers are shown at each node, percentage genetic distance scale is shown below the tree. Two DHDPS clades can be identified: DHDPS A-type and DHDPS B-type, the latter being legume-specific and the former present in all plant species including legumes. DHDPS sequences of *G. max* are boxed (Gm.DHDPS-A1; Glyma.09G268200, GmDHDPS-A2; Glyma.18G221700 and Gm.DHDPS-B; Glyma.03G022300).

**Figure 3 plants-11-01762-f003:**
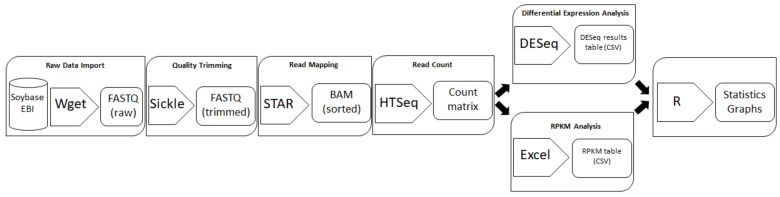
Schematic representation of the bioinformatics pipeline and workflow for high-throughput RNA-seq data re-analysis used in this paper. Each raw publicly available FASTQ file is downloaded from the database using the command line utility Wget, FASTQ files are trimmed on quality using Sickle, reads are mapped on the reference genome Wm82.a2.v1 and sorted into BAM files by STAR, HTSeq counts the gene expression reads per gene into an HTSeq-count matrix, count matrices per experiment serve as input for DESeq analysis or can be normalized by calculating the RPKM, finally DESeq result tables or RPKM tables can be further statistically analyzed or graphically visualized using R.

**Figure 4 plants-11-01762-f004:**
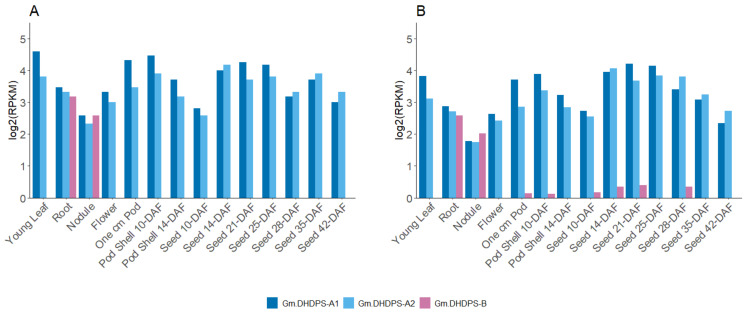
Bar chart comparison between RPKM values (*Y*-axis) of the three *DHDPS* genes in 14 different plant samples (*X*-axis) as (**A**) publicly available at https://soybase.org/, accessed on 21 June 2022 and (**B**) the RPKM values obtained by re-analysis [31]. All RPKM values are log2-transformed.

**Figure 5 plants-11-01762-f005:**
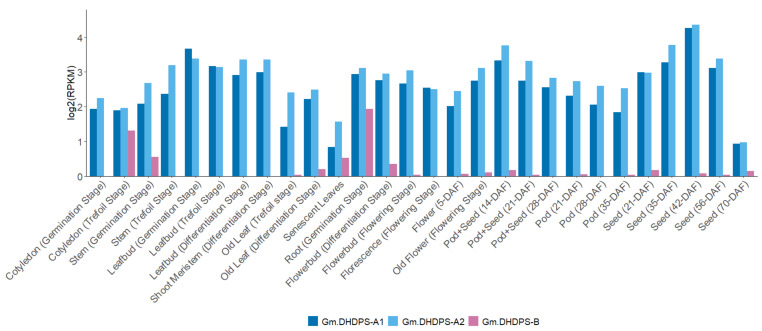
Bar chart representation of the *DHDPS* RPKM values (*Y*-axis) in 28 samples (*X*-axis) in *G. max* by RNA-seq re-analysis of data originating from [32]. All RPKM values are log2-transformed; (DAF = Days After Flowering).

**Figure 6 plants-11-01762-f006:**
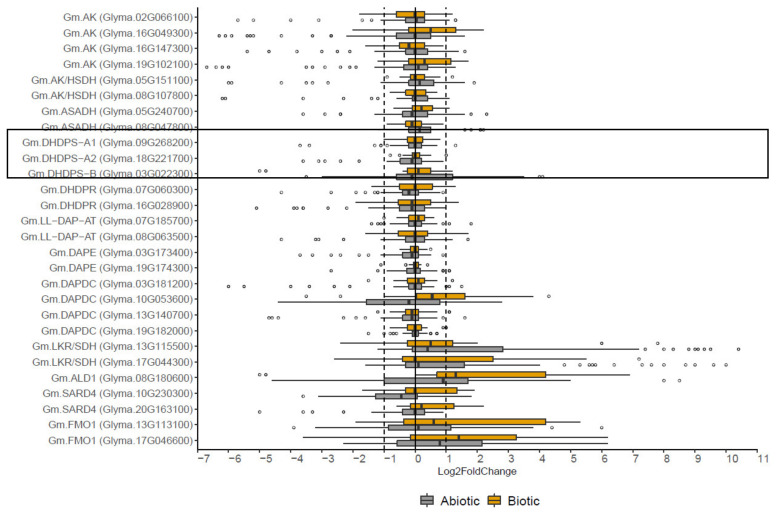
Boxplots for differential expression values (Log2Fold changes, *X*-axis) of genes (*Y*-axis) involved in lysine biosynthesis (*AK, AK/HSDH, ASADH, DHDPS* (boxed)*, DHDPR, LL-DAP-AT, DAPE, DAPDC*), lysine catabolism (*LKD/SDH*) and SAR response (*ALD1, SARD4, FMO1*) by re-analysis of 23 biotic and 70 abiotic stress RNA-seq experiments in soybean.

**Figure 7 plants-11-01762-f007:**
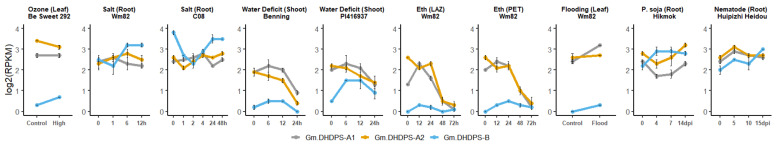
Line graphs of log2-transformed RPKM values for *DHDPS* for re-analyzed experiments in which at least one *DHDPS* was significantly up- or downregulated (*p* < 0.05). Each graph title consists in short of the treatment, plant tissue and genotype of the experiment. *Y*-axis represents the RPKM value and each *X*-axis the treatment in chronological or alphabetical order. Eth = ethylene; LAZ = leaf abscission zone; PET = petiole after LAZ removed.

**Table 1 plants-11-01762-t001:** Amino acid substitution table adapted from [19] for specific sites in DHDPS known for catalytic activity and lysine inhibition allosteric sites for *Physcomitrella patens (Pp)*, *Selaginella moellendorffii (Sm)*, *Picea abies (Pa)*, *Oryza sativa (Os)*, *Zea mays (Zm)*, *Sorghum bicolor (Sb)*, *Aquilegia coerulea (Ac)*, *Vitis vinifera (Vv)*, *Eucalyptus grandis (Eg)*, *Arabidopsis thaliana (At)*, *Populus trichocarpa (Pt)*, *Glycine max (Gm)*, *Vigna unguiculata (Vu)*, *Phaseolus vulgaris (Pv)*, *Pisum sativum (Ps)*, *Medicago truncatula (Mt)* and *Lotus japonicus (Lj)*, with amino acid position numbering from *N. sylvestris* DHDPS [20]. *G. max* protein names are underlined. The consensus amino acid sequence is based on the multiple alignments of 156 legume DHDPS protein sequences (see Materials and Methods). The amino acid color scheme was adapted from [30], with each color representing different physicochemical characteristics of the amino acids.

	Sites Known For Catalytic Activity				Allosteric Sites (Lysine Inhibition)	
	68	69	130	131	155	160	183	185	204	206	207	222	261	73	77	80	81	102	103	104	108	111	112
**CONSENSUS**	T	T	Y	Y	Y	R	K	C	G	D	D	I	N	Q	W	H	I	G	S	N	E	H	A
**Pp.DHDPS-A1**	*	*	*	*	*	*	*	*	*	*	*	*	*	*	*	*	*	*	*	*	*	*	*
**Pp.DHDPS-A2**	*	*	*	*	*	*	*	*	*	*	*	*	*	*	*	*	*	*	*	*	*	*	*
**Sm.DHDPS-A**	*	*	*	*	*	*	*	*	*	*	*	*	*	H	*	*	*	*	*	*	*	*	*
**Pa.DHDPS-A1**	*	*	*	*	*	*	*	*	*	*	*	*	*	*	*	*	*	*	*	*	*	*	*
**Pa.DHDPS-A2**	*	*	*	*	*	*	*	*	*	*	*	*	*	*	*	*	*	*	*	*	*	*	*
**Sb.DHDPS-A1**	*	*	*	*	*	*	*	*	*	*	*	*	*	H	*	*	*	*	*	*	*	*	*
**Sb.DHDPS-A2**	*	*	*	*	*	*	*	*	*	*	*	*	*	H	*	*	*	*	*	*	*	*	*
**Zm.DHDPS-A1**	*	*	*	*	*	*	*	*	*	*	*	*	*	H	*	*	*	*	*	*	*	*	*
**Zm.DHDPS-A2**	*	*	*	*	*	*	*	*	*	*	*	*	*	H	*	*	*	*	*	*	*	*	*
**Oz.DHDPS-A1**	*	*	*	*	*	*	*	*	*	*	*	*	*	H	*	*	*	*	*	*	*	*	*
**Oz.DHDPS-A2**	*	*	*	*	*	*	*	*	*	*	*	*	*	H	*	*	*	*	*	*	*	*	*
**Ac.DHDPS-A1**	*	*	*	*	*	*	*	*	*	*	*	V	*	H	*	*	*	*	*	*	*	*	*
**Ac.DHDPS-A2**	*	*	*	*	*	*	*	*	*	*	*	V	*	H	*	*	*	*	*	*	*	*	*
**Vv.DHDPS-A**	*	*	*	*	*	*	*	*	*	*	*	*	*	*	*	*	*	*	*	*	*	*	*
**Pt.DHDPS-A1**	*	*	*	*	*	*	*	*	*	*	*	*	*	*	*	*	*	*	*	*	*	*	*
**Pt.DHDPS-A2**	*	*	*	*	*	*	*	*	*	*	*	V	*	*	*	*	*	*	*	*	*	*	*
**Eg.DHDPS-A1**	*	*	*	*	*	*	*	*	*	*	*	*	*	*	*	*	*	*	*	*	*	*	*
**Eg.DHDPS-A2**	A	V	*	*	*	*	*	*	*	*	Y	M	T	H	*	*	*	*	R	*	*	Q	*
**Eg.DHDPS-A3**	*	A	*	*	*	*	*	*	*	*	*	M	*	H	*	*	*	*	*	*	*	E	*
**Eg.DHDPS-A4**	*	A	*	*	*	*	*	*	*	*	H	M	A	H	*	*	*	*	*	*	*	Q	*
**AtDHDPS2**	*	*	*	*	*	*	*	*	*	*	*	*	*	*	*	*	*	*	*	*	*	*	*
**AtDHDPS1**	*	*	*	*	*	*	*	*	*	*	*	*	*	*	*	*	*	*	*	*	*	*	*
**Lj.DHDPS-A**	*	*	*	*	*	*	*	*	*	*	*	*	*	*	*	*	*	*	*	*	*	*	*
**Mt.DHDPS-A1**	*	*	*	*	*	*	*	*	*	*	*	*	*	*	*	*	*	*	*	*	*	*	*
**Mt.DHDPS-A2**	*	*	*	*	*	*	*	*	*	*	*	*	*	*	*	*	*	*	*	*	*	*	*
**Ps.DHDPS-A**	*	*	*	*	*	*	*	*	*	*	*	V	*	*	*	*	*	*	*	*	*	*	*
**Pv.DHDPS-A**	*	*	*	*	*	*	*	*	*	*	*	*	*	*	*	*	*	*	*	*	*	*	*
**Vu.DHDPS-A**	*	*	*	*	*	*	*	*	*	*	*	*	*	*	*	*	*	*	*	*	*	*	*
** Gm.DHDPS-A1 **	*	*	*	*	*	*	*	*	*	*	*	*	*	*	*	*	*	*	*	*	*	*	*
** Gm.DHDPS-A2 **	*	*	*	*	*	*	*	*	*	*	*	V	*	*	*	*	*	*	*	*	*	*	*
**Lj.DHDPS-B1**	S	*	*	*	*	*	*	G	*	*	K	Q	T	*	*	Q	*	*	*	*	*	*	*
**Lj.DHDPS-B2**	*	*	*	*	*	*	*	*	A	*	E	M	T	*	L	K	*	*	*	*	Q	S	I
**Lj.DHDPS-B3**	*	*	*	*	*	*	*	Y	S	E	K	M	T	*	V	K	V	*	*	*	Q	S	I
**Lj.DHDPS-B4**	*	*	*	*	*	*	*	*	A	Q	R	M	I	*	L	K	*	*	*	*	Q	S	I
**Mt.DHDPS-B1**	*	*	*	*	*	*	*	*	*	*	E	H	A	Y	*	Q	*	*	*	*	*	T	*
**Mt.DHDPS-B2**	S	*	*	*	*	*	*	*	-	*	I	Q	S	*	*	Q	*	*	*	*	*	N	*
**Mt.DHDPS-B3**	S	*	*	*	*	*	*	*	-	*	I	Q	S	*	*	Q	*	*	*	*	*	N	*
**Mt.DHDPS-B4**	*	*	*	*	*	*	*	*	A	Q	K	V	I	H	L	K	V	*	*	*	L	S	L
**Ps.DHDPS-B1**	*	*	*	*	*	*	*	*	*	*	E	H	T	Y	*	Q	*	*	*	*	*	T	*
**Ps.DHDPS-B2**	S	*	*	*	*	*	*	*	-	*	G	Q	S	*	*	Q	*	*	*	*	*	E	*
**Ps.DHDPS-B3**	*	*	*	*	*	*	*	*	A	*	E	M	C	*	I	K	V	*	*	*	Q	T	I
**Pv.DHDPS-B**	S	*	*	*	*	*	*	*	*	*	K	Q	V	*	*	Q	*	*	*	*	*	K	*
**Vu.DHDPS-B**	S	*	*	*	*	*	*	*	*	*	K	Q	G	*	*	Q	*	*	*	*	*	K	*
** Gm.DHDPS-B **	*	*	*	*	*	*	*	*	*	*	K	Q	V	*	*	Q	*	*	*	*	*	K	*

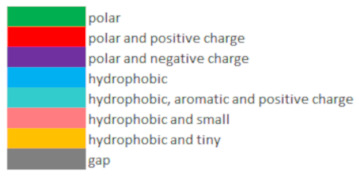

**Table 2 plants-11-01762-t002:** Comparison between the original gene expression atlas data and the re-analysis using the bioinformatics pipeline from this paper [31]. Total reads and % uniquely mapped reads per plant sample are given. In addition, the Pearson’s correlation coefficient was calculated for all gene expression counts within each sample. (*** = *p* < 0.0001).

	Total Reads		Unique Reads (%)		Correlation
Sample	Severin et al.	This Paper	Severin et al.	This Paper	(Pearson’s r)
young leaf	6,618,852	6,621,825	66%	72%	0.991 ***
flower	5,829,223	5,800,039	60%	67%	0.962 ***
one cm pod	6,181,917	6,149,218	58%	64%	0.977 ***
pod shell 10DAF	6,464,386	6,426,175	58%	63%	0.992 ***
pod shell 14DAF	5,983,354	5,921,485	50%	57%	0.909 ***
seed 10DAF	6,962,047	6,936,823	44%	47%	0.999 ***
seed 14DAF	5,888,849	5,845,764	43%	47%	0.999 ***
seed 21DAF	2,711,453	2,692,219	39%	44%	0.985 ***
seed 25DAF	7,835,063	7,142,607	37%	48%	0.989 ***
seed 28DAF	9,673,118	8,010,459	26%	35%	0.991 ***
seed 35DAF	9,102,649	8,791,274	52%	64%	0.997 ***
seed 42DAF	7,052,993	6,884,047	49%	60%	0.998 ***
root	8,402,716	8,402,716	57%	68%	0.933 ***
nodule	8,930,860	8,930,860	61%	68%	0.995 ***

**Table 3 plants-11-01762-t003:** Significant differential expression values only (DESeq Wald test, * = *p* < 0.05, ** = *p* < 0.01, *** = *p* < 0.001) for *DHDPS* only from the re-analyzed 23 biotic and 70 abiotic stress experiments in soybean. SRA Study ID, soybean genotype, experiment within the SRA study, tissue type, enzyme ID, Log2Fold change, and base mean (mean over control and treatment samples) are given. Cut-off values for Log2Fold change values were set to +1 for upregulated genes and −1 for downregulated genes.

SRA Study	Genotype, Experiment, Tissue	Enzyme	Log2Fold Change	Base Mean
SRP050050	Wm82, Ethylene 12 h, Leaf Abscission Zone	Gm.DHDPS-A1	1.3 *	27.9
SRP155375	Himok, *P. soja* 4 dpi, Root	Gm.DHDPS-A1	−1 *	166.6
SRP050050	Wm82, Ethylene 48 h, Leaf Petiole	Gm.DHDPS-A1	−1.1 *	23.5
SRP045932	Benning, Water Deficit 24 h, Shoot	Gm.DHDPS-A1	−1.1 ***	95.7
SRP050050	Wm82, Ethylene 72 h, Leaf Abscission Zone	Gm.DHDPS-A1	−3.4 *	5.1
SRP050050	Wm82, Ethylene 72 h, Leaf Petiole	Gm.DHDPS-A1	−3.7 ***	10.2
SRP050050	Wm82, Ethylene 48 h, Leaf Petiole	Gm.DHDPS-A2	−1.8 ***	30.7
SRP045932	Benning, Water Deficit 24 h, Shoot	Gm.DHDPS-A2	−2.4 ***	65.1
SRP050050	Wm82, Ethylene 48 h, Leaf Abscission Zone	Gm.DHDPS-A2	−2.9 ***	17.9
SRP050050	Wm82, Ethylene 72 h, Leaf Petiole	Gm.DHDPS-A2	−3.1 ***	15.8
SRP050050	Wm82, Ethylene 72 h, Leaf Abscission Zone	Gm.DHDPS-A2	−3.6 ***	15.3
SRP050050	Wm82, Ethylene 48 h, Leaf Petiole	Gm.DHDPS-B	4.1 *	1.8
SRP050050	Wm82, Ethylene 24 h, Leaf Petiole	Gm.DHDPS-B	4.0 *	1.6
SRP076153	Wm82, Flooding, Leaf	Gm.DHDPS-B	2.7 *	5.6
SRP045932	PI416937, Water Deficit 12 h, Shoot	Gm.DHDPS-B	1.8 **	54.4
SRP045932	Benning, Water Deficit 12 h, Shoot	Gm.DHDPS-B	1.7 *	12.4
SRP009826	Be Sweet 292, Ozone, Leaf	Gm.DHDPS-B	1.5 ***	26.0
SRP045932	PI416937, Water Deficit 24 h, Shoot	Gm.DHDPS-B	1.3 *	30.5
SRP135932	Heidou, Nematode 15 dpi, Root	Gm.DHDPS-B	1.2 ***	316.0
SRP041622	Wm82, Salt 12 h, Root	Gm.DHDPS-B	1.0 ***	148.0
SRP132150	C08, Salt 2 h, Root	Gm.DHDPS-B	−1.7 ***	350.8

## Data Availability

All original RNA-seq data used for the re-analysis in this paper is publicly available at https://www.ebi.ac.uk/ena/browser/home, accessed 1 December 2020) as filed under the experiments SRP025919, SRP038111, SRP009826, SRP024277, SRP031889, SRP035871, SRP041622, SRP045932, SRP050050, SRP076153, SRP058975, SRP064384, SRP105922-SRP105965, SRP132150, SRP108540, SRP155375, SRP056137, SRP091708, SRP126743 and SRP135932.

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
