# Peer review of "The Hidden Potential of High-Throughput RNA-Seq Re-Analysis, a Case Study for DHDPS, Key Enzyme of the Aspartate-Derived Lysine Biosynthesis Pathway and Its Role in Abiotic and Biotic Stress Responses in Soybean"

_plants, 2022, doi:10.3390/plants11131762_

Round 1
Reviewer 1 Report
The authors investigated the biological relevance of DHDPS in soybean by first conducting a phylogenetic anaylsis, which identified the common DHDPSs and legume specific DHDPSs. Using a collection of multiple sets of public RNA-seq data, the authors then investigated the expression patterns of soybean DHDPS genes in the contexts of different tissues, abiotic- and biotic stress conditions. While these results give insights to the functional aspects of DHDPSs in soybean for further study, the authors need to refine the analysis steps and revised the texts accordingly.
Major comments:
1. Due to the shape of the trees in Figure 2, it is recommended to change it to a circular tree for better visulaisation when reading in a page.
2. In Figure 4, it is not clear whether the y-axis represent "RPKM" or "log2(RPKM)". Because the y-axis label is "RPKM" but the authors wrote that "All RPKM values are log2 transformed" in the figure legend.
3. At line 539, the authors should indicate the version of PlantCyc database.
4. The authors peroformed a validation for the RNA-seq analysis method in terms of the expression patterns in dataset. For the subsequent datastes for comparing treatment and control samples, the authors should also perform quality control steps to check for any con-founding factors within the same dataset. For example, PCA can be performed to check whether the samples were clustered according to the biological replicates and conditions. This analysis will indicate whether any corrections are needed before the authors can use the expression levels of certain genes. Other other hand, the authors should also check whether the fold-change of genes in a dataset distribute in a reasonable way by looking at the volcano plot. Because when the authors observed that there were very little changes in the DHDPS genes, it could be that the changes were indeed very small, or most genes were not differentially expressed due to some unknown issues of the dataset. After these quality control steps, the authors can decided whether to perform corrections for the dataset or just exclude it. Since the manuscript focuses on RNA-seq analysis, it is important to have strict quality control steps. The results of these quality control steps can be presented in the supplementary documents.
5. For RNA-seq data analysis, although the authors only focused on the expression level of the DHDPS genes, corrections for multiple differential expression testing are sitll necessary. Hence, at line 580, the authors should mention about using the adjusted p-value from DESeq2.
Minor comments:
1. The authors should carefully check the typos and gramma throughout the texts.
Author Response
Major comments:
- Due to the shape of the trees in Figure 2, it is recommended to change it to a circular tree for better visulaisation when reading in a page.
Response 1: The shape of the tree was changed to radial and is now aesthetically sound.
2. In Figure 4, it is not clear whether the y-axis represent "RPKM" or "log2(RPKM)". Because the y-axis label is "RPKM" but the authors wrote that "All RPKM values are log2 transformed" in the figure legend.
Response 2: I put log2(RPKM) in all Y-axes (Figure 4, Figure 5 and Figure 7) as it is indeed log2 transformed RKPM values and now more clear for the reader.
3. At line 539, the authors should indicate the version of PlantCyc database.
Response 3: adjusted to PlantCyc (v15.0)
4. The authors peroformed a validation for the RNA-seq analysis method in terms of the expression patterns in dataset. For the subsequent datastes for comparing treatment and control samples, the authors should also perform quality control steps to check for any con-founding factors within the same dataset. For example, PCA can be performed to check whether the samples were clustered according to the biological replicates and conditions. This analysis will indicate whether any corrections are needed before the authors can use the expression levels of certain genes. Other other hand, the authors should also check whether the fold-change of genes in a dataset distribute in a reasonable way by looking at the volcano plot. Because when the authors observed that there were very little changes in the DHDPS genes, it could be that the changes were indeed very small, or most genes were not differentially expressed due to some unknown issues of the dataset. After these quality control steps, the authors can decided whether to perform corrections for the dataset or just exclude it. Since the manuscript focuses on RNA-seq analysis, it is important to have strict quality control steps. The results of these quality control steps can be presented in the supplementary documents.
4. Response: I totally agree with this comment. Volcano plots in addition to a PCA analysis were added for all abiotic and biotic stress experiments used in the paper and implemented in Figure S3.The experiments filtered for significant differentially expressed DHDPS genes showed no abnormal vulcano plots. However volcano plots for experiments SRP024277 (Ozone treatment), SRP024277 (Drought), SRP031889 (Fe deficiency), SRP058975 (water deficiency), SRP064384 (CO2 and drought) showed a relatively low amount of significant differentially expressed genes implying less chance to find DHDPS to be differentially expressed as well in these experiments. In one experiment, being SRP056137 (Fusarium oxysporum infection), the volcano plot of the infection with the non-pathogenic strain at 96 hpi (hours post infection) is very irregular and results from this DESeq analysis should be excluded for further analysis.
Following the PCA analysis most samples for all experiments clustered accordingly except for some problems with SRP108540 and SRP056137. However these PCA clustering anomalies do not affect the final expression analysis and results of the DHDPS genes are DHDPS was not found to be significant differentially expressed in experiments with irregular volcano or inconsistent PCA clusters as mentioned now at line 323-325.
This quality control information was added to the result section and to the material and methods. Lines 290-301 and 600 to 602, respectively.
5. For RNA-seq data analysis, although the authors only focused on the expression level of the DHDPS genes, corrections for multiple differential expression testing are sitll necessary. Hence, at line 580, the authors should mention about using the adjusted p-value from DESeq2.
Response 5: Adjusted in the Material and Methods. Indeed we used the adjusted p-value to correct for multiple testing.
Minor comments:
- The authors should carefully check the typos and gramma throughout the texts
Resonse 1: Will do a final read for typos and grammatical errors before resubmission.
PS I attached the Supplemental file with the extra PCA and Volcano plots.

Reviewer 2 Report
The manuscript “The hidden potential of high-throughput RNA-seq re-analysis, a case study for DHDPS, key enzyme of the aspartate-derived lysine biosynthesis pathway and its role in abiotic and biotic stress responses in soybean” performed the phylogenetical analysis of DHDPS in soybean and elucidated their expression profile in different tissue samples and biotic and abiotic treatment based on published transcriptome data. Here are some comments.
L207 “up to 12% more unique reads in comparison with the original read mapping (Table 2).”, 12% might be 72%.
Figure 7 might be related with Table 3 in the context. However, I didn’t find the connect between “air exposure” in Figure 7 and “water deficit” in Table 3.
Author Response
L207 “up to 12% more unique reads in comparison with the original read mapping (Table 2).”, 12% might be 72%.
Response 1: Indeed typo, corrected
Figure 7 might be related with Table 3 in the context. However, I didn’t find the connect between “air exposure” in Figure 7 and “water deficit” in Table 3.
Response 2: Indeed! The linked is fixed and I adjusted Figure 7 to be in sync with Table 3.
Round 2
Reviewer 1 Report
The authors have addressed the previous comments. Since the authors have surveyed a large number of datasets and identify those not passing QC by volcano plot analysis and PCA, it is better that plots of the datasets passing the QC are shown in one supplementary figure and those not passing are shown in another figure. This can be a nice reference for audience who would like to use those datasets. Also, it is better to adjust the width and length of those PCA and volcano plots, in order to make it consistent.